# A Reproducibility Study of Differentially Private Fine-Tuning of Diffusion Models

## Abstract

Differentially private (DP) fine-tuning of generative models has emerged as a promising direction for adapting powerful pretrained systems to sensitive domains without compromising individual privacy. In this work, we investigate whether the qualitative findings of DP-LoRA Tsai et al. (2025) remain reproducible under a practical backbone substitution and realistic compute constraints through an independent empirical reproduction and extension study for high-resolution conditional image synthesis, using CelebA-HQ and a publicly pretrained latent diffusion backbone. We find that qualitative DP-LoRA trends remain stable for the substituted backbone evaluated in this study, while absolute utility metrics are affected by the choice of pretrained backbone. Our results reproduce the core behavioral trends: image quality improves as the privacy budget relaxes, moderate LoRA ranks and thoughtful adapter placement matter, and parameter-efficient fine-tuning stays around 1% of total model parameters across all configurations. We assess reproducibility in terms of qualitative trends rather than exact numerical replication. Beyond reproduction, we extend the study with additional ablations on DP-SGD hyperparameters, static rank allocation schedules, alternating LoRA schedules and a localized DP feasibility analysis. While our absolute metrics differ from the original, the qualitative behavior of DP-LoRA remains consistent. Our code is available at: `https://github.com/anonymous2026-5-1/dp-lora-reproducibility`.

## 1 Introduction

With the rapid growth of deep generative models, which often require vast amounts of data for training, concerns regarding data privacy have become increasingly prominent (Duan et al., 2023). This has led to research on privacy-preserving learning methods aimed at preventing the disclosure of sensitive training data. Among these approaches, Differential Privacy (DP) (Dwork, 2006) has emerged as a simple yet effective framework that provides a mathematical guarantee for privacy protection. Incorporating DP into the design of a deep generative model mitigates the risk of reproducing sensitive training data from the model parameters or inference outputs.

Diffusion Models (DMs) are widely used for image generation due to their ability to produce high-quality outputs. However, their high memorization capacity leads to privacy concerns, such as the leaking of training data. Prior work Duan et al. (2023) has demonstrated that both pixel-space and latent diffusion models can be vulnerable to membership-inference attacks (MIAs), which could become a problem when training with sensitive data, such as medical images. Tsai et al. (2025) propose DP-LoRA, a parameter-efficient method for improving the privacy of latent diffusion models (LDMs) by applying DP. They first pre-train an LDM on a public, non-sensitive dataset followed by fine-tuning using a private dataset with Low-Rank Adaptation (LoRA) (Hu et al., 2022) and differentially-private stochastic gradient descent (DP-SGD) (Abadi et al., 2016).

Reproducibility is particularly important for differentially private generative models because privacy guarantees often come at significant utility costs and computational expense. DP-LoRA is a representative parameter-efficient approach for private diffusion model adaptation, yet reproducing its findings is challenging because the original pretrained checkpoints are not publicly available. While benchmarking efforts such

as DPImageBench (Gong et al., 2025) evaluate DP-LoRA alongside other private synthesis methods, they do not specifically investigate reproducibility under backbone substitution. We therefore investigate whether the central qualitative conclusions of DP-LoRA remain valid when the original pretrained checkpoints are unavailable.

We make the following contributions in this study:

- Evaluating whether the qualitative findings of DP-LoRA remain reproducible under a practical backbone substitution.
- Extending the original study with ablations on DP-SGD hyperparameters, static layer-wise LoRA rank allocation schedules and alternating LoRA training schedules.
- Presenting a feasibility study of localized differential privacy for autoencoders.

## 2  Background

**Differential Privacy**  DP (Dwork, 2006) provides a formal guarantee that the output of a randomized mechanism does not change significantly when a single individual's data is modified or removed. A mechanism $\mathcal{M}$ satisfies $(\epsilon, \delta)$-differential privacy if, for all neighboring datasets $D$ and $D'$ and all measurable output sets $S$, it holds that

$$\Pr[\mathcal{M}(D) \in S] \leq \exp(\epsilon) \cdot \Pr[\mathcal{M}(D') \in S] + \delta. \tag{1}$$

The guarantee given by equation 1 ensures that the outputs $\mathcal{M}(D)$ and $\mathcal{M}(D')$ are similar in distribution, making it difficult for an adversary to determine whether or not an entry was included in the dataset used to calculate the result. This difficulty is determined by the privacy budget $\epsilon$ and the failure probability $\delta$, where smaller values for them provide a stronger guarantee. In the context of diffusion models, the goal is to ensure that it is difficult for a user of a trained model (the adversary) to infer the images used for training the model.

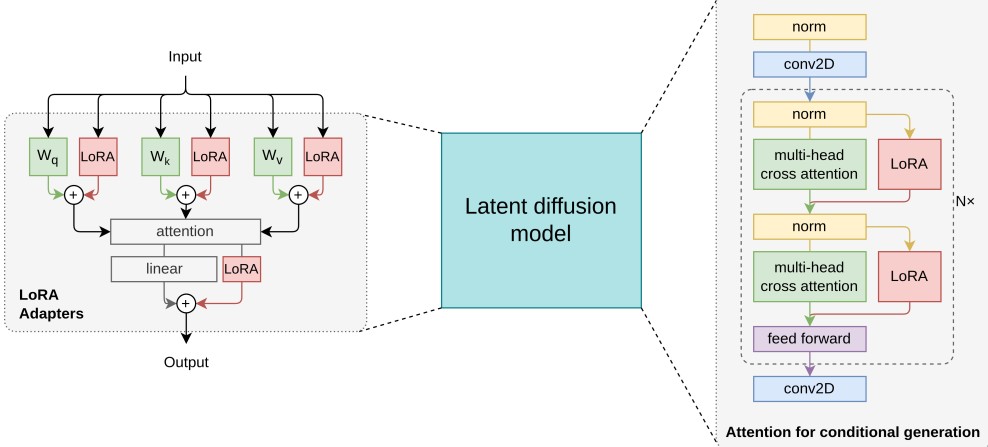

Figure 1: The placement of the LoRA adapters within an LDM.

**DP Optimization**  We use stochastic gradient descent with differential privacy (DP-SGD) (Abadi et al., 2016) which is a widely used optimization technique for incorporating DP in Deep Learning. However, alternative optimization algorithms like DP-Adam also exist (McMahan et al., 2019). These methods implement DP through clipping gradients using a clipping norm $C$, and adding noise to those gradients. The micro-batch size, amount of training iterations and noise variance all determine the final values for $\epsilon$ and $\delta$.

**Differentially-Private LoRA for LDMs**  LDMs provide a lower dimensionality latent space in contrast to regular DMs. However, even with low-dimensionality, they remain expensive to fine-tune under DP when the number of parameters grows too large. DP-LoRA addresses this challenge by introducing low-rank

adapters to attention layers and the output projection layer as shown in Figure 1, while freezing the original weights of the model. This not only reduces the amount of parameters, but also decreases the amount of noise injected during DP-SGD. This enables stable and efficient private fine-tuning for a wide variety of private datasets and resolutions.

**Localized DP**  In many real-world applications, only localized regions of an image contain sensitive information (e.g., faces, identifiers, or medical markers). In such settings, standard DP-SGD can be overly conservative, as it assumes a worst-case sensitivity of the model parameters that is independent of the actual data being used, as discussed in Thudi et al. (2024). This overestimation significantly degrades model utility. Consequently, this motivates *Localized DP*, where DP noise is added to updates most influenced by a sensitive region-of-interest, with the goal of preserving utility compared to globally privatizing the full model.

## 3  Related Work

**Privacy attacks on generative models**  The vulnerability of generative models to privacy attacks has been studied extensively. Carlini et al. (2023) showed that diffusion models can memorize and reproduce verbatim training samples under certain conditions, particularly when images appear multiple times in training data. Duan et al. (2023) extended this to membership inference attacks (MIAs), demonstrating that both pixel-space and latent diffusion models are susceptible to an adversary trying to determine whether a particular image was used during training. These findings motivate the use of DP as a formal defense, since DP-SGD provides a provable bound on how much any individual training example can influence the model output.

**Differentially private deep learning**  DP-SGD (Abadi et al., 2016) is the canonical method for training neural networks with DP guarantees. It clips per-sample gradients and adds calibrated Gaussian noise at each step. While effective, DP-SGD is known to degrade model utility significantly, especially in high-dimensional parameter spaces, since the noise required to achieve a meaningful privacy budget scales with the number of parameters. This has led to a line of work focused on reducing the effective parameter count under DP. Yu et al. (2022) showed that fine-tuning only a small adapter on top of a frozen pretrained model substantially improves the privacy-utility trade-off compared to full fine-tuning, since fewer parameters are exposed to gradient noise. DP-LoRA builds directly on this insight by combining LoRA with DP-SGD. An alternative to DP-SGD is DP-Adam (McMahan et al., 2019), which incorporates adaptive gradient estimates, though its interaction with LoRA under DP has received less study.

**DP fine-tuning for diffusion models**  Several works have explored DP training specifically for generative image models. DP-LDM Liu et al. (2024) fine-tunes a latent diffusion model under DP-SGD but operates on a much larger trainable parameter set ($\approx$10% of total parameters), resulting in higher noise injection and worse utility under the same privacy budget. Ghalebikesabi et al. (2023) propose pre-training on public data followed by DP fine-tuning on private data, a strategy that DP-LoRA also adopts, reinforcing its effectiveness.

DPImageBench (Gong et al., 2025) provides a recent benchmark for comparing differentially private image synthesis methods. We cite it primarily to contextualize the broader DP image generation landscape rather than for direct quantitative comparison, since the architectures, datasets, and evaluation protocols differ substantially from DP-LoRA. Unlike our work, DPImageBench evaluates DP-LoRA as one method within a broader benchmark and does not investigate whether the qualitative findings of the original paper remain reproducible under a different backbone.

Recent methods such as PrivImage (Li et al., 2024) and DP-FETA (Gong et al., 2026) further improve private image synthesis and are included in recent benchmark studies such as DPImageBench. Unlike these approaches, which focus on private image generation itself, DP-LoRA studies parameter-efficient private adaptation of pretrained diffusion models.

**Parameter-efficient fine-tuning** LoRA (Hu et al., 2022) is one of the most widely used parameter-efficient fine-tuning (PEFT) methods. It reparameterizes weight updates as low-rank matrix products, drastically reducing the number of trainable parameters without significantly affecting expressivity for many tasks. AdaLoRA (Zhang et al., 2023) extends this by adaptively allocating rank budgets across layers during training based on a learned importance criterion, achieving better parameter efficiency than uniform-rank LoRA. In our study we evaluate a simpler static variant of non-uniform rank allocation (E7), since data-dependent rank updates complicate DP accounting. Other PEFT approaches such as prefix tuning Li & Liang (2021) and adapter layers Houlsby et al. (2019) have also been explored in the DP context, though LoRA has emerged as particularly well-suited due to its minimal parameter overhead and straightforward integration with gradient clipping.

**Localized and structured privacy** Standard DP-SGD treats all parameters symmetrically, which can be unnecessarily wasteful when sensitive information is spatially concentrated. Several works have explored structured or input-dependent privacy. **?** analyzed the trade-off between privacy and representation learning, showing that DP can disproportionately hurt features that are rare or fine-grained. Gradient-based sensitivity analysis (Thudi et al., 2024) suggests that not all gradients carry equal private information, motivating selective noise injection. Our RoI-only DP study (E8) is in the same spirit: by restricting noise to parameters most activated by the sensitive image region, we aim to preserve utility for non-sensitive parts of the reconstruction. This connects to the broader idea of input-level or feature-level DP, though a fully rigorous treatment of such localized guarantees remains an open research problem.

## 4 Scope of Reproducibility

We aim to reproduce the claimed behavior reported in Tsai et al. (2025) instead of exact numerical results due to the unavailability of the original pretrained LDMs and the computational costs involved in pretraining on large datasets like ImageNet. We therefore evaluate reproducibility in one practical backbone-substitution setting rather than robustness across multiple architectures. We empirically focus on fine-tuning and evaluating on a publicly available pretrained LDM (`cin256`) (Rombach et al., 2022) and the CelebA-HQ dataset (Karras et al., 2018) for conditional high-resolution image synthesis. We selected `cin256` because it is one of the few publicly available ImageNet-pretrained LDM checkpoints whose architecture is sufficiently compatible with the original DP-LoRA implementation. This allows us to assess DP-LoRA under practical constraints and isolate the effect of the backbone while avoiding re-implementation of the methodology that could otherwise confound the study. We also initialize an LDM for the (E)MNIST dataset (Cohen et al., 2017) at a resolution of $32 \times 32$ to confirm the parameter counts reported in the original work.

We investigate the following claims from the original paper in our study:

- **Claim 1**: Privacy-Utility trade-off (at high resolution)
  Holding all other hyperparameters fixed, relaxing the privacy budget improves image quality. In particular, FID scores decrease as $\epsilon$ increases over $\{1, 5, 10\}$. Consequently at $\epsilon = \infty$, we get no privacy guarantee, but the best FID score.
- **Claim 2**: Ablation trends
  At $\epsilon = 10$, (i) a moderate LoRA rank achieves the best FID-parameter count trade-off among $r \in \{8, 16, 32\}$, (ii) increasing noise multiplicity steps improves FID at the cost of longer epoch times (we test $k \in \{1, 2, 4\}$ due to memory limits, skipping $k = 8$), and (iii) placing adapters in both attention (QKV) and projection layers yields the best FID than placing adapters in either component alone.
- **Claim 3**: Parameter Efficiency
  DP-LoRA fine-tunes with a very small trainable subset of parameters (approximately $\leq 1\%$ of total parameters) while maintaining competitive FIDs under $\epsilon \leq 10$.

In addition to these claims, we extend the original work to investigate the feasibility and impact of the following on training stability and privacy-utility trade-off:

- **Ablation study** of DP-SGD hyperparameters – Clipping norm $C$ and DP micro-batch size – to localize impact on utility.
- **Feasibility study** Static layer-wise rank allocation, following either a linear or cosine decay applied at initialization to test impact of depth-dependent parameter allocation on utility.
- **Feasibility study** Alternating LoRA training (alternating updates to A and B matrices).
- **Feasibility study** Localized DP: formulating and evaluating the practicality of privatizing only designated regions of an image (e.g., the eyes in face datasets) under DP-SGD for Autoencoders.

## 5 Methodology

We use the authors' released code-base and modify it to run in an environment with our specific experimental setup. A detailed breakdown of model architectures and other details regarding reproducibility can be found in Appendix A. Table 1 shows the key differences between the original and our setup.

| Experimental aspect | DP-LoRA | Our study |
| --- | --- | --- |
| Pretrained backbone | Original LDM | Public `cin256` LDM |
| Dataset | CelebA-32/64 | CelebA-HQ |
| Image resolution | $32^2$ / $64^2$ | $256^2$ |
| FID samples | 50,000 | 10,000 |
| Primary comparison | Absolute FID | Qualitative trends |

Table 1: Key experimental differences between DP-LoRA and our reproducibility study.

### 5.1 Model Description

We use the publicly available Class-conditional LDM by Rombach et al. (2022). It is pretrained on ImageNet-1k (Russakovsky et al., 2015). The model consists of two parts: a Vector-Quantized Autoencoder that up-/down-samples images from $256 \times 256$ to $32 \times 32$ and back, and the UNet denoising network that operates on a 32-dimensional space. Model configurations can be found in Table 14.

### 5.2 Datasets

We primarily use CelebA-HQ, which is a high resolution ($256 \times 256$) dataset, to match our objective of evaluating high-resolution image synthesis, as provided on *HuggingFace* Karras et al. (2018). It has a total of 30,000 images, with gendered class labels (male/female) for conditional generation. It has a train/validation split of 28,000/2,000. The data is center-cropped and normalized to a $[-1, 1]$ range, as is standard for diffusion models. Although the pretrained backbone was originally trained using ImageNet class conditioning, we replace the conditioning labels during fine-tuning with the binary gender labels provided by CelebA-HQ. The class-conditioning embeddings are adapted accordingly while the remaining backbone remains unchanged. In our configuration, the class-conditioning embedding is trainable and is included in the optimizer passed to Opacus. Therefore, it is also covered by the DP-SGD accounting.

### 5.3 Hyperparameters

Unless otherwise stated, all experiments use a learning rate of $1 \times 10^{-6}$, train for 10 epochs with random seed 23, LoRA rank $r = 8$, clipping norm $C = 5 \times 10^{-4}$, DP micro-batch size 8, noise multiplicity $k = 2$, and $\epsilon = 10$. Complete training and DP configurations are provided in Appendix A.

### 5.4 Evaluation metrics

**Fréchet Inception Distance (FID)**   We use FID as our primary evaluation metric. FID is computed using the standard Inception-V3 network between 10,000 generated images and the full CelebA-HQ dataset (30,000 images). Compared to the original DP-LoRA evaluation, which uses 50,000 generated samples, a

different pretrained backbone, and a different CelebA variant, our protocol has higher estimator variance and is not directly numerically comparable. We therefore interpret FID as evidence of qualitative trends rather than exact numerical replication.

**Parameter counts**  To evaluate parameter efficiency (**Claim 3**), we report trainable parameter counts for all models. For each model, only LoRA components contribute to trainable parameters, since the backbone remains frozen during fine-tuning.

## 5.5  Experiments Overview

Table 2 summarizes all our experiments. Each experiment (E0-E7) varies a single factor while keeping all other settings fixed according to Table 16. We fine-tune and evaluate FID for each experiment.

| ID | Experiment | Varied setting | Claim |
|----|------------|----------------|-------|
| E0 | Baseline | No DP ($\epsilon = \infty$) | 1, 3 |
| E1 | Privacy budget | $\epsilon \in \{1, 5, 10\}$ | 1 |
| E2 | LoRA rank | $r \in \{8, 16, 32\}$ | 2, 3 |
| E3 | Adapter placement | QKV / Proj / Both | 2, 3 |
| E4 | Noise multiplicity | $k \in \{1, 2, 4\}$ | 2 |
| E5 | DP clipping norm | $C \in \{2.5, 5, 10\} \times 10^{-4}$ | Ext. |
| E6 | DP micro-batch size | $\{4, 8, 16\}$ | Ext. |
| E7 | Rank scheduling | $16 \rightarrow 4/8$ (lin./cos.) | Ext. |
| E8 | Localized DP-LoRA | No/Global/Localized DP for Autoencoder | Ext. |
| E9 | Alternating LoRA | Alternating A/B matrix updates | Ext. |

Table 2:  Overview of experimental configurations. Unless stated otherwise, all experiments use the default hyperparameters in Table 16 and are evaluated using FID on 10,000 generated samples.

## 5.6  Extended Methods

**Static Layer-wise Rank Allocation (E7)**  We evaluate non-uniform LoRA ranks by decaying the rank from 16 to either 4 or 8 using linear or cosine schedules across layer depth (Appendix A). The objective is to allocate greater capacity to earlier layers while reducing parameter count in deeper layers.

**Alternating LoRA Training (E9)**  Standard LoRA trains both the $A$ and $B$ adapter matrices simultaneously. In E9, we test an alternating schedule: in odd epochs, only $A$ is updated while $B$ is frozen, and in even epochs, only $B$ is updated while $A$ is frozen. This approach is motivated by the idea that decoupling the two matrices may reduce the effective gradient sensitivity per step, which could help control DP noise. The privacy accounting still treats both matrices as trainable parameters. We evaluate this against the standard simultaneous training baseline using FID at $\epsilon = 10$.

**Localized DP (E8).**  We investigate whether differential privacy can be localized to parameters most influenced by a sensitive region-of-interest (RoI). Using eye-region masks from CelebAMask-HQ, we identify a subset of autoencoder parameters associated with the RoI through activation analysis and apply DP-SGD only to this subset while updating all remaining parameters normally. The selected subset covers approximately 6% of autoencoder parameters. Since this experiment is scoped to the autoencoder rather than the full latent diffusion model, utility is evaluated using downstream gender-classification accuracy on latent representations instead of FID. Additional implementation details, privacy assumptions, and limitations are provided in Appendix C. Localized DP should not be interpreted as providing a full-model differential privacy guarantee.

| Finding | DP-LoRA | This work |
|---|---|---|
| Higher $\epsilon$ improves utility | Reported | Reproduced |
| Moderate LoRA rank is preferred | $r = 16$ | $r = 16$ |
| QKV + Projection performs best | Reported | Reproduced |
| Higher noise multiplicity improves FID | Reported | Reproduced |
| ∼1% trainable parameters | ∼1% | ∼1% ($r = 32$: 1.07%) |

Table 3: Comparison between the qualitative findings reported by DP-LoRA and the observations reproduced in this work. Absolute FID values are not compared because the pretrained backbone, dataset, image resolution, and evaluation protocol differ.

## 6 Results

Table 3 summarizes the key results. Absolute FIDs are worse than those reported in the original paper, which we attribute to differences in pretrained backbones, the datasets used and compute constraints. We treat replication success in terms of qualitative trends, rather than exact numerical replication.

### 6.1 Results reproducing original paper

**Claim 1: Privacy-Utility Trade-off** Table 4 shows the FID scores obtained on the ablations of privacy budget. We see that the FID scores decrease as we increase the privacy budget. When fine-tuned without DP, the privacy guarantee is removed, which immediately improves utility. This model is used as our baseline.

| $\varepsilon$ | FID ↓ | Params (Train/Total) |
|---|---|---|
| 1 | 149.28 | 1.25M / 464.47M |
| 5 | 62.40 | 1.25M / 464.47M |
| 10 | **50.28** | 1.25M / 464.47M |
| ∞ | 22.41 | 1.25M / 464.47M |

Table 4: (E0-E1) FID scores and parameter counts of the standard configurations at different values for $\epsilon$. $\epsilon = \infty$ indicates the no-DP baseline.

| $r$ | FID ↓ | Params (Train/Total) |
|---|---|---|
| 8 | 50.28 | 1.25M / 464.47M |
| 16 | **37.92** | 2.50M / 465.72M |
| 32 | 37.65 | 5.00M / 468.22M |

Table 5: (E2) FID scores and parameter counts for different ranks $r$ at $\epsilon = 10$

**Claim 2: Ablation trends** Table 5 shows FID scores for different values of LoRA rank $r$. The rank has a large impact on the amount of trainable parameters, which scales linearly with $r$. Setting $r = 16$ seems to give the best trade-off between the FID score and the parameter counts, which matches the original paper.

Table 6 shows the FID scores for different LoRA adapter placements. It indicates that placing the LoRA adapters in both the QKV and Projection layers yields the best FID scores. It also shows that LoRA adapters placed only in projections layers perform significantly worse than those placed only in attention layers (178.35 compared to 64.26). Additionally, we also see that placing them in both layers yields the best parameter-efficiency.

| FT Modules | FID ↓ | Params (Train/Total) |
|---|---|---|
| QKV only | 64.26 | 0.93M / 464.16M |
| Projection only | 178.35 | 0.32M / 463.54M |
| QKV + Projection | **50.28** | 1.25M / 464.47M |

Table 6: (E3) FID scores and parameter counts for different LoRA adapter placements at $\epsilon = 10$

| $k$ | FID ↓ | Train time (min per epoch) |
|---|---|---|
| 1 | 83.67 | 21 |
| 2 | 50.28 | 22 |
| 4 | **31.39** | 35 |

Table 7: (E4) FID scores and training time per epoch for different noise multiplicity values $k$

| Dataset | Configuration | Trainable Params | Total Params | % Trainable |
|---------|---------------|-----------------:|-------------:|------------:|
| MNIST | Default | 56,880 | 107,285,894 | 0.053 |
| CelebA-HQ | Base | 1,249,280 | 464,474,771 | 0.269 |
| | $r = 16$ | 2,498,560 | 465,724,051 | 0.537 |
| | $r = 32$ | 4,997,120 | 468,222,611 | 1.067 |
| | Projection-only | 319,488 | 463,544,979 | 0.069 |
| | QKV-only | 929,792 | 464,155,283 | 0.200 |

Table 8: (E0,E2,E3) Trainable and total parameter counts

Table 7 shows the effect of the noise multiplicity value $k$ on the FID scores. Note that the value $k = 8$ has not been examined due to memory constraints. Increasing the noise multiplicity results in a lower FID score but it also increases training time per epoch, which matches the trends of the original paper.

**Claim 3: Parameter Efficiency**    Table 8 shows the parameter counts of the fine-tuned models for each dataset. The parameter count for the model initialized on (E)MNIST matches the reported parameters in the original paper. For the remaining models the ratio of trainable parameters remains around 1% across all configurations, with only $r = 32$ slightly exceeding this threshold, since the rank $r$ has the largest impact on the trainable parameter count.

## 6.2    Extensions

**Ablation study of DP-SGD hyperparameters**    We varied the clipping norm C and DP micro-batch size to evaluate their effect on utility. Table 9 shows that, under our single-seed evaluation, varying $C$ produced little observable change in utility. This suggests that the gradients in our LoRA setup may already lie in a space where modest changes in $C$ do not translate into quality differences.

| Clipping norm $C$ | FID ↓ |
|-------------------|-------|
| $2.5 \times 10^{-4}$ | 50.2060 |
| $5.0 \times 10^{-4}$ | 50.2824 |
| $1.0 \times 10^{-3}$ | 50.1238 |

Table 9: (E5) Ablation on DP-SGD clipping norm $C$. Utility is largely unchanged across this range.

| DP micro-batch size | FID ↓ |
|---------------------|-------|
| 4 | – |
| 8 | 50.2824 |
| 16 | 47.8295 |

Table 10: (E6) Ablation on DP micro-batch size. Larger DP micro-batches slightly improve FID but are memory-limited.

Increasing the DP micro-batch size from 8 to 16 improved FID (Table 10) slightly. However we were not able to test batch size 32 due to memory constraints. Meanwhile, batch size 4 failed early in our setup likely due to an inconsistency between how the code splits the batch for DP micro-batching and how attention/conditioning tensors are constructed, hence we report it as infeasible in our setup.

**Static Layer-wise Rank Allocation**    Table 11 shows the FID scores of depth-wise rank allocation with linear and cosine decay compared to fixed-rank baselines. We see that, while the number of parameters reduces drastically compared to $r = 16$, the performance drops closer to the lower-rank ($r = 8$) baseline in both cases. However, decay from $16 \rightarrow 8$ does improve performance marginally over fixed-rank $r = 8$.

**Alternating LoRA Training**    Table 13 compares simultaneous vs. alternating LoRA updates at $\epsilon = 10$. Alternating training resulted in slightly worse FID than standard simultaneous training. We attribute this to slower convergence: alternating updates effectively halve the learning signal for each matrix per epoch, which may be especially harmful under DP noise. The alternating schedule did not yield any meaningful reduction in privacy cost either, since both matrices remain in the accountant's trainable parameter set regardless of whether they are updated in a given epoch.

| Rank setting | FID ↓ | Params (Train/Total) |
|---|---|---|
| 8 (fixed) | 50.28 | 1.25M / 464.47M |
| 16 (fixed) | **37.92** | 2.50M / 465.72M |
| $16 \rightarrow 4$ (lin.) | 47.25 | 1.61M / 464.84M |
| $16 \rightarrow 4$ (cos.) | 50.22 | 1.62M / 464.86M |
| $16 \rightarrow 8$ (lin.) | 42.06 | 1.91M / 465.13M |
| $16 \rightarrow 8$ (cos.) | 42.48 | 1.92M / 465.18M |

Table 11: (E7) FID scores and parameters counts for different rank settings at $\epsilon = 10$.

| Training Config | Accuracy | Noised Param(%) |
|---|---|---|
| Non-DP | 92.4 | - |
| Global DP | 87.2 | 100% |
| Localized DP | 91.3 | 6% |

Table 12: (E8) Downstream gender classification accuracy scores on CelebA-HQ (at $\epsilon = 10$, where applicable).

| Update scheme | FID ↓ |
|---|---|
| Simultaneous (standard) | **50.28** |
| Alternating A/B | 75.14 |

Table 13: (E9) FID comparison of simultaneous vs. alternating LoRA updates at $\epsilon = 10$.

**Localized DP** Table 12 compares No-DP, Global DP, and Localized DP on downstream gender-classification accuracy. Localized DP perturbs only ≈6% of parameters (those most activated by the eye region mask) yet achieves accuracy close to the No-DP setting. Global DP, which adds noise to all parameters, suffers a larger accuracy drop under the same budget. This suggests that spatially localized privatization may preserve utility in tasks where sensitive information is spatially concentrated.

## 7 Discussion

**Claim 1: Privacy-Utility Trade-off** – Supported

Our experiments reproduce the expected privacy–utility trade-off: increasing $\epsilon$ consistently improves image quality, matching the qualitative behavior reported by DP-LoRA. Although absolute FID differs because of backbone and evaluation differences, the trend remains consistent. The backbone substitution may have a larger effect on absolute FID than any individual DP-LoRA hyperparameter explored in this study. Our non-private baseline remains worse than the best results in the original paper. This suggests that pretrained backbone choice may influence absolute utility more strongly than the individual DP-LoRA hyperparameters explored in this study.

**Claim 2: Ablation trends** – Partially Supported

Our experiments reproduce the qualitative ablation trends reported in the original paper. A moderate LoRA rank (r=16), higher noise multiplicity, and joint attention-projection adaptation consistently produced the best results. However, computational constraints prevented a full quantitative replication.

**Claim 3: Parameter efficiency** – Supported

Our parameter counts confirm that DP-LoRA remains highly parameter efficient, with trainable parameters remaining around 1% across evaluated configurations. DP-LoRA enables effective fine-tuning with a substantially smaller parameter subset than prior DP methods, such as DP-LDM which requires approximately 10% of the parameters, as reported by Liu et al. (2024).

**Extension Studies** Under our single-seed evaluation, varying the clipping norm produced little observable change in FID, while larger DP micro-batches produced a small improvement in our experiments. This is likely due to the averaging of noise over several gradients that stabilizes the updates. Static rank allocation reduced parameter counts but did not outperform fixed-rank LoRA. Our results suggest that in our CelebA-HQ setup, preserving a uniformly higher LoRA capacity across layers is more beneficial. Alternating LoRA

updates degraded performance, likely due to slower convergence under DP noise, and provided no privacy benefit.

**Localized DP**   Our Localized DP results indicate that restricting DP noise to parameters most influenced by sensitive regions may preserve utility compared to Global DP under the same privacy budget. In E8, Localized DP achieves accuracy close to the No-DP setting while perturbing only a small fraction of parameters, whereas Global DP introduces a larger utility drop.

These results indicate that localized privatization may be a promising direction for tasks where sensitive information is localized spatially. However, E8 is a feasibility study, that does not currently propose a complete privacy framework. Stronger guarantees depend on how RoI selection and masking are defined and accounted for in the DP analysis. Additionally, our work has not been extended to the full LDM, scoped down to only the autoencoder. Localized DP is best viewed as a utility-preserving heuristic inspired by localized sensitivity rather than a complete replacement for standard DP-SGD.

**Limitations**

- Computational resources were limited for this study. Only two GPUs were available with limited memory. Although these GPUs are powerful, the number of GPU hours available in combination with a lack of available pretrained checkpoints was not enough to reproduce the results exactly.
- The checkpoints for the LDMs pre-trained on ImageNet were not available. This meant that they needed to be trained from scratch, which was infeasible for ImageNet given the computational resources and time.
- Localized DP assumptions: The effectiveness of Localized DP depends on reliable RoI segmentation and on the pretrained model's exposure to the sensitive features; mismatches may weaken both utility and privacy guarantees.
- All experiments were conducted using a single random seed due to computational constraints. Consequently, the reported numerical values should be interpreted as evidence of qualitative trends rather than statistically significant differences. Repeated runs and uncertainty estimates remain important future work.
- We do not include qualitative image samples because the substituted pretrained backbone produces visually inferior generations compared to the unavailable original checkpoints, making qualitative visual comparison potentially misleading. Our evaluation therefore focuses on reproducing the reported quantitative trends using the original paper's primary metric (FID).

**Broader Impact**   Differentially private fine-tuning can reduce the risk of memorizing sensitive training data while enabling generative models to be adapted to privacy-sensitive domains such as healthcare and biometrics. By independently reproducing and extending DP-LoRA, our work contributes to improving confidence in privacy-preserving generative modeling and highlights the practical impact of implementation choices such as pretrained backbone selection.

Our Localized DP feasibility study should not be interpreted as a complete differential privacy mechanism for released models. Rather, it is intended as a feasibility study exploring whether selectively privatizing subsets of parameters can better preserve utility. We explicitly discuss its limitations and encourage future work to develop rigorous privacy guarantees. More broadly, we hope our released implementation and transparent discussion of reproducibility challenges support more reliable and reproducible research in differentially private machine learning.

## 8   Conclusion

The goal of this study was to reproduce and extend the results presented by Tsai et al. (2025). Overall, our findings reproduce the qualitative trends of the original work. While computational constraints prevented a complete replication of the experiments, the observed trends are in line with those from the original paper, even when using a different pre-trained model and dataset. This suggests that the reported qualitative behavior persists in the backbone-substitution setting evaluated in this study.

Our extension studies found limited sensitivity to clipping norm, modest gains from larger DP micro-batches, no benefit from alternating LoRA updates, and encouraging utility preservation from Localized DP. These findings provide additional evidence regarding the practical behavior of DP-LoRA under constrained settings. Our results suggest that the principal qualitative conclusions reported by DP-LoRA remain reproducible in the practical backbone-substitution setting evaluated in this study, while absolute utility metrics remain sensitive to backbone choice and evaluation protocol.

Future work could include, (i) exploring DP-safe dynamic rank allocation strategies for LoRA adapters, for example, by modifying a method such as *AdaLoRA* (Zhang et al., 2023), which adaptively allocates LoRA ranks during training, (ii) extending and establishing a Localized DP paradigm for the full LDM, and (iii) evaluating reproduced models using empirical privacy attacks such as membership inference and reconstruction attacks to complement formal differential privacy guarantees.

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

# A    Reproducibility

## A.1    Code setup and compute

The code is used as-is from the original authors with some environment-specific changes including fixing dependencies and creating quality-of-life documentation and scripts that are documented in Appendix A.

To measure the environmental impact of training the models, *CodeCarbon* (Courty et al., 2024) is integrated into the code. It records runtime and power usage for each hardware component (CPU, GPU, and RAM) used during training and calculates how much energy was consumed and carbon emitted, based on the region provided. In our case, we make use of *Snellius*, a supercomputer based in Amsterdam, The Netherlands.

To run the code, two different GPUs were used: NVIDIA A100 with 40 GB of RAM and a Multi Instance GPU (MIG) NVIDIA A100 with 20 GB of RAM. The A100 GPU was only used in case the MIG partition did not have enough memory available.

### A.2 Reproducibility Notes

The code was published by the authors, so our study intentionally uses the released implementation rather than an independent reimplementation. This allows observed differences to be attributed primarily to the substituted pretrained backbone and experimental setup rather than implementation differences. We document all modifications made to the original code and release them together with our implementation for transparency and reproducibility.

The primary issue when reproducing the original paper was insufficient computational resources. To reproduce the results exactly, it is necessary to pre-train models on ImageNet, which is a large dataset. This would require many GPU-hours to complete, which was not feasible with our computational and time constraints. Because of this, we opted to use a pre-trained model called `cin256`. This model has been trained on ImageNet, but it has a slightly different architecture than the model used by the authors.

While reproducing the paper, we attempted to contact the original authors of the paper to request checkpoints of the ImageNet-pretrained autoencoders and LDMs, since training them from scratch was infeasible. However, the authors did not reply to our request. We also sent a request for checkpoints to the authors of a different paper who used DP-LoRA in a comparison study with other DP methods (Gong et al., 2025). They replied quickly and did send two checkpoints. These had a smaller model architecture than used in the original DP-LoRA paper. However, we opted not to use these in favor of `cin256`.

**Changes in code**  The original code was cloned into our repository, available on *GitHub*. We made several changes to allow the code to run correctly in our environment:

- Adding the missing data loader for EMNIST - the original code had configurations that referred to a data loader for EMNIST that did not exist.

- Adding installation guides for the pretrained model and dataset required by our experiments.

- Writing an installation script for the environment - using the provided environment file did not install the required packages correctly.

- Writing fine-tuning configuration files to match `cin256` and include the correct parameters for our experiments.

- Adjusting and refactoring the code to include the layer-wise rank allocation with linear/cosine schedules as defined below:

$$r_{\text{lin}} = r_{\text{init}} + (r_{\text{final}} - r_{\text{init}}) \cdot \ell$$
$$r_{\text{cos}} = r_{\text{final}} + \tfrac{1}{2}(r_{\text{init}} - r_{\text{final}})\left(1 + \cos(\pi \cdot \ell)\right)$$

- Implementing CodeCarbon in the relevant files for fine-tuning and sampling.

- Implementing Localized DP finetuning for Autoencoders.

### A.3 Model Configurations

Below we detail the model configurations used in our specific setup.

| Attribute | cin256 | | EMNIST pre-trained LDM | |
|---|---|---|---|---|
| | *AutoencoderKL* | *UNet* | *AutoencoderKL* | *UNet* |
| Image Size | 256 | 32 | 32 | 4 |
| Latent size | 32 | 8 | 4 | 2 |
| Channels (in/out) | 3/3 | 4/4 | 3/3 | 3/3 |
| Channels (latent) | 4 | 256 | 3 | 128 |

Table 14: Model configurations for `cin256` and (E)MNIST-initialized latent diffusion models.

## A.4 Complete DP Configuration

Table 15 details the exact DP configurations/parameters as used in the original code with minimal changes to extend to our selected backbone.

| Item | Value |
|---|---|
| Optimizer | AdamW with DP-SGD via Opacus |
| Accountant | Opacus PrivacyEngine |
| Opacus version | 1.4.0 |
| RDP accountant | No; Poisson setting uses PRV calibration |
| Poisson sampling | Yes |
| Sampling rate | $q = 1/\lceil 27000/2048 \rceil \approx 0.0714$ |
| Noise multiplier $\sigma$ | Calibrated at runtime for target $\epsilon$ using Opacus PRV |
| Clipping norm | $5 \times 10^{-4}$ |
| Batch size | 2048 logical batch size |
| Microbatch / physical batch | 8 via virtual batch splitting |
| Optimizer steps | $\approx 14$ per epoch, $\approx 140$ total |
| Epochs | 10 |
| $\delta$ | $10^{-5}$ |
| $\epsilon$ | 10 |
| Trainable DP parameters | UNet LoRA parameters plus class-conditioning embeddings |
| Class-conditioning embeddings | Trainable and included in DP accounting |

Table 15: Differentially private fine-tuning configuration for CelebA-HQ.

## A.5 Full training configurations

All full configuration files used for experiments are provided in our repository (`dp_lora/reproducibility_experiments/`). We include the configuration files for all ablation variants (privacy budget, rank, adapter placement, $k$, clipping norm $C$, DP micro-batch size, rank schedules), and each includes the relevant configuration for the relevant model architecture used. Below are some of the default hyperparameters used across experiments unless stated otherwise:

## A.6 Hardware used per experiment

Due to queue availability and memory constraints, experiments were run either on a full NVIDIA A100 (40 GB) or on a 20 GB A100 MIG partition. Table 17 lists the hardware used per fine-tuning run. All sampling runs were done on A100 (40 GB).

| Type | Parameter | Value |
|---|---|---|
| Optimization | Learning rate | $1 \times 10^{-6}$ |
| | Max epochs | 10 |
| | Random seed | 23 |
| DP | $\epsilon$ | 10 |
| | $\delta$ | $1 \times 10^{-5}$ |
| | $C$ | 0.0005 |
| | Micro-batch size | 8 |
| | Noise steps $k$ | 2 |
| LoRA | Rank $r$ | 8 |
| | Target | projection + output |
| Data / Eval | Data batch size | 2048 |
| | Conditioning | `class_label` |
| | FID samples | 10,000 |

Table 16: Default hyperparameters (unless stated otherwise).

| Experiment | Setting | GPU (fine-tune) | GPU (sampling) |
|---|---|---|---|
| E0,E1 | $\epsilon \in 1, 5, 10, \infty$ | MIG | A100 |
| E2 | $r \in \{8, 16\}$ | MIG | A100 |
| E2 | $r = 32$ | A100 | A100 |
| E3 | (QKV/Proj/Both) | MIG | A100 |
| E4 | $k \in 1, 2, 4$ | A100 | A100 |
| E5 | $C \in \{2.5, 5, 10\} \times 10^{-4}$ | MIG | A100 |
| E6 | 8 | MIG | A100 |
| E6 | 16 | A100 | A100 |
| E7 | $16 \rightarrow 4/8$ (lin./cos.) | MIG | A100 |
| E8 | $\epsilon \in 1, 10$ | A100 | A100 (Only generation) |

Table 17: GPU hardware used per experiment.

## B Environmental Impact Calculations and Runtimes

Across configurations, fine-tuning a DP-LoRA model for 10 epochs on CelebA-HQ required approximately **7-8 hours**, while conditional sampling of 10,000 images for FID evaluation required an additional **2-3 hours** per run.

In total, our reproducibility study consumed approximately **170 GPU-hours** of compute time. Using *CodeCarbon* measurements and extrapolating them we estimate that a total of about **60 kWh** of energy was used for our experiments and about **16 kgCO$_2$** was emitted. These estimates are intended to provide an indication of environmental cost rather than exact measurements.

Due to the unavailability of EAR (Snellius' Energy Aware Runtime) and limited compute access, direct environmental measurements were only obtained for a subset of experiments using CodeCarbon, indicated with an asterisk (*) in the below table. These were used to estimate per-GPU-hour energy consumption and CO$_2$ emissions, which were then extrapolated to the remaining experiments based on their total wall-clock runtime. Therefore, experiments in Table 18 reflect extrapolated CodeCarbon estimates, not direct measurements. Water usage could not be estimated due to the absence of WUE data in both CodeCarbon and the Snellius environment.

*Note that higher the noise multiplicity $k$, the higher the runtime, and vice versa. This is because higher noise multiplicity introduces more forward-backward passes through the LDM per epoch. This also means that it pulls more power and emits more CO$_2$.*

| Config | Total (h) | Energy (kWh) | $CO_2$ (kg) |
|---|---|---|---|
| $\epsilon = 1$ (E1) | 10.440 | 3.790 | 1.014 |
| $\epsilon = 5$ (E1) | 10.437 | 3.789 | 1.014 |
| $\epsilon = 10^*$ (E1) | 10.429 | 3.271 | 0.874 |
| $\epsilon = \infty^*$ (E0) | 9.409 | 2.886 | 0.772 |
| $r = 16$ (E2) | 10.459 | 3.796 | 1.016 |
| $r = 32$ (E2) | 10.486 | 3.806 | 1.019 |
| $k = 1^*$ (E4) | 5.541 | 1.905 | 0.617 |
| $k = 2$ (E4: ft-only)* | 3.739 | 1.405 | 0.376 |
| $k = 4$ (E4) | 8.714 | 3.462 | 0.927 |
| qkvonly (E3) | 10.121 | 3.678 | 0.984 |
| projonly (E3) | 9.548 | 3.481 | 0.932 |
| $C = 0.00025$ (E5) | 10.419 | 3.783 | 1.012 |
| $C = 0.001$ (E5) | 10.429 | 3.786 | 1.013 |
| dp-batch = 16 (E6) | 6.051 | 2.461 | 0.659 |
| Linear $16 \rightarrow 4^*$ (E7) | 10.480 | 3.804 | 1.018 |
| Cosine $16 \rightarrow 4^*$ (E7) | 10.468 | 3.792 | 1.015 |
| Linear $16 \rightarrow 8^*$ (E7) | 10.472 | 2.843 | 0.761 |
| Cosine $16 \rightarrow 8^*$ (E7) | 10.472 | 3.229 | 0.864 |
| Classification (E8)* | 0.25 | 0.127 | 0.041 |
| Autoencoder fine-tuning (E8)* | 1.5 | 0.616 | 0.165 |

Table 18: Total runtime for fine-tuning and sampling, and extrapolated environmental impact per experiment, using CodeCarbon. Values are rounded to three decimal places. (*) indicates experiments for which CodeCarbon was run.

## C Localized DP for Autoencoders

### C.1 Mask construction details

The eye-region masks for CelebA-HQ were sourced from the CelebAMask-HQ dataset, which provides $512 \times 512$ pixel-level semantic annotations for 19 facial attributes. We merge the `l_eye` and `r_eye` label channels into a single binary mask and resize it to $256 \times 256$ using nearest-neighbour interpolation to match the input resolution. No dilation or morphological post-processing was applied. Each mask is loaded per image at training time via `feature_path`. The mask is applied before the forward pass by zeroing pixels outside the RoI, with outside pixels set to the channel mean rather than zero to avoid hard-boundary artefacts: $\tilde{x} = x \odot m + \bar{x} \odot (1 - m)$, where $\bar{x}$ is the per-channel mean.

### C.2 Sensitivity threshold selection

The 75th-percentile activation threshold ($\tau = Q_{75}$) was chosen to keep the noised parameter fraction small ($\approx 6\%$) while still covering the filters most strongly activated by eye-region content. The threshold is computed once over a representative batch of 512 masked images before training begins and is fixed for the duration of fine-tuning. In principle, $\tau$ is a hyperparameter that trades off utility preservation (lower $\tau \Rightarrow$ more parameters noised $\Rightarrow$ approaches Global DP) against coverage of RoI-sensitive computation (higher $\tau \Rightarrow$ fewer parameters noised $\Rightarrow$ weaker spatial coverage). We verified that the selected filters correspond to early-to-mid encoder layers, consistent with the expectation that low-level texture and shape features are encoded at shallow depths.

### C.3 Privacy accounting and assumptions

The formal $(\epsilon, \delta)$-DP guarantee in E8 applies only to the parameters in $\mathcal{S}$. The Opacus RDP accountant runs on this reduced parameter set under the same noise multiplier $\sigma$ and sampling rate as Global DP, yielding

$\epsilon = 10$ but over a much smaller slice of the model. Parameters outside $\mathcal{S}$ receive no formal guarantee. This rests on two assumptions that our feasibility study does not fully verify:

1. **Activation separability.** Parameters not selected by the threshold are assumed to encode negligible information about the RoI. This is a reasonable approximation when the RoI is spatially compact and receptive fields are well-localized, but it does not hold strictly for deep layers where receptive fields cover much of the image. Such layers may still carry residual RoI signal even when their activation norm under $\tilde{x}$ falls below $\tau$.

2. **Fixed mask independence.** The selection set $\mathcal{S}$ is computed before training and held fixed. If it were recomputed data-adaptively at each step, this would consume additional privacy budget and would need to be accounted for via mechanisms such as report-noisy-max or a private threshold. Our static selection avoids this, but means $\mathcal{S}$ may drift from the true sensitive parameter set as the autoencoder adapts during training.

A complete Localized DP framework would need to account for both the mask selection procedure and residual sensitivity of unmasked parameters, for example following the sensitivity-weighted noise allocation of Andrew et al. (2021). We leave this formalization to future work.

### C.4 Implementation constraints and scope

E8 operates on the autoencoder only, not the full LDM. Two practical reasons forced this scope. First, the UNet's attention layers aggregate spatial information globally, making it difficult to attribute any filter activation exclusively to the RoI without more sophisticated gradient attribution. Second, Opacus' per-sample gradient hooks interact poorly with the cross-attention conditioning pathway in our codebase, causing silent shape mismatches when partial parameter masking is applied. The convolutional autoencoder avoids both issues: gradients flow through locally receptive layers, making the activation-based selection more interpretable and the binary masking straightforward to apply at the optimizer step.

Downstream gender-classification accuracy serves as the utility proxy because FID requires full end-to-end image generation through the LDM. The ResNet-18 classifier is trained on the frozen autoencoder's latent codes using identical hyperparameters across all three conditions (Non-DP, Global DP, Localized DP); only the autoencoder checkpoint differs. Instructions for mask download, classifier training, and running E8 end-to-end are in the repository `README`.

