# OpenReview forum: "A Reproducibility Study of Differentially Private Fine-Tuning of Diffusion Models"
_TMLR — Under review for TMLR_

### Review · Reviewer_Xuwn · 2026-07-06

**Summary Of Contributions:**

This paper presents an independent reproducibility and extension study of DP-LoRA for differentially private fine-tuning of latent diffusion models. Because the original pretrained checkpoints are unavailable, the authors evaluate whether the qualitative findings of DP-LoRA persist under a different publicly available latent diffusion backbone and practical compute constraints. The paper studies high-resolution conditional image generation on CelebA-HQ and reports that the main qualitative trends are preserved: image quality improves as the privacy budget increases, moderate LoRA ranks give a better utility/parameter trade-off, adapting both attention and projection layers performs best, higher noise multiplicity improves FID at increased training cost, and DP-LoRA keeps the trainable parameter fraction around or below 1% in most settings.

The paper also adds several extension experiments beyond reproduction: ablations on the DP-SGD clipping norm and DP micro-batch size, static layer-wise rank schedules, alternating LoRA matrix updates, and a feasibility study of RoI-localized DP for autoencoders. The strengths of the work are its practical reproducibility focus, its clear acknowledgement of compute and checkpoint limitations, its useful ablations, and its attempt to separate qualitative reproducibility from exact numerical replication. The paper is also transparent about hardware and environmental costs.

The main weaknesses are that the evidence is still limited: most claims are based on one substituted backbone, one main dataset, apparently one seed, and no uncertainty estimates. This makes the term “robustness” somewhat stronger than the experiments justify. The RoI-only DP experiment is interesting but currently not a full DP guarantee for the released model and should be presented more carefully. Several DP accounting and evaluation details also need to be made more explicit before the reader can fully assess the strength of the privacy and reproducibility claims. In addition, the paper writing is not in a good format of publication, e.g. no need to include Table 2 for hyperparameters in main paper, authors might move most of the content from section 5 to appendix.

**Audience:**

Yes

**Audience Explanation:**

A subset of the TMLR audience would likely be interested in this paper, especially researchers working on differentially private deep learning, private generative modeling, diffusion models, and reproducibility. DP fine-tuning of large generative models is computationally expensive and sensitive to pretrained backbones, so a careful report showing which DP-LoRA trends survive under a different public backbone is useful. The paper also provides practical information about compute limitations, implementation issues, parameter counts, and environmental costs, which can help other researchers attempting similar reproductions.

**Broader Impact Concerns:**

The paper studies privacy-preserving training of generative models, so the broader impact is directly relevant. I did not see a dedicated broader impact statement in the main paper. I recommend adding one.

**Claims And Evidence:**

No

**Claims Explanation:**

The central qualitative claims are mostly supported by the reported experiments. The privacy-utility trend is clear: FID improves as ε increases, and the no-DP baseline gives the best FID. The adapter placement and noise multiplicity trends are also reasonably supported by the reported tables. The parameter-efficiency claim is supported by the parameter counts, with the caveat that the r=32 setting slightly exceeds 1% trainable parameters.

However, the evidence is not yet fully convincing for the broader claims about robustness and reproducibility. The paper replaces the original backbone with a single public backbone and evaluates primarily on CelebA-HQ, so the results show that the trends persist in one substituted setting, not that they are broadly robust to backbone substitution. The absence of multiple random seeds, confidence intervals, or FID variance estimates makes it difficult to tell whether smaller differences, such as rank-scheduling variants or clipping-norm ablations, are meaningful. The paper also uses 10,000 generated samples for FID rather than the 50,000 samples used in the original work, which is understandable for compute reasons but weakens direct comparability.

The RoI-only DP section is the largest source of concern. The paper appropriately notes that this is not a full-model DP guarantee, but some phrasing and table labels could still mislead readers into interpreting it as a valid localized differential privacy mechanism for the released autoencoder or downstream model. Since parameters outside the selected set receive no formal guarantee, this should be framed as a heuristic feasibility study unless the authors provide a rigorous privacy statement and accounting for the full released mechanism.

Overall, the main empirical trends are plausible and useful, but several claims should be narrowed or supported with stronger evidence.

**Requested Changes:**

- Narrow the robustness claims or add more evidence. The current experiments use one substituted backbone and one main high-resolution dataset. The authors should either evaluate at least one additional backbone/dataset or revise the title, abstract, and conclusion to say that the study evaluates one practical backbone-substitution setting rather than robustness in general.
- Clarify the FID protocol and comparability to the original paper. The paper should more directly compare its evaluation protocol to the original DP-LoRA protocol, including the number of generated samples, train/test/reference split used for FID, resolution, dataset differences, and backbone differences. If 10k-sample FID is used for compute reasons, the authors should explicitly discuss how this affects variance and comparability.
- Provide complete DP accounting details. The authors should report the noise multipliers, sampling rates, number of optimization steps, batch construction, micro-batch handling, clipping norm, accountant settings, and exactly which parameters are included in the DP mechanism in appendix. They should also clarify whether class-conditioning embeddings or other non-LoRA components are trained and, if so, whether they are covered by DP-SGD.
- Reframe the RoI-only DP experiment. The RoI-only experiment should not be presented as providing full DP for the released autoencoder or generated outputs. The authors should either provide a rigorous privacy formulation for the full mechanism or clearly rename/reframe this as “localized noise injection” or “partial-parameter DP feasibility study.” The table should avoid suggesting that the full model has ε=10 DP.
- Include original-paper numbers side-by-side where possible. Since this is a reproducibility study, the reader needs a clear mapping between the original claims and the reproduced results. A table with original DP-LoRA results, this paper’s corresponding results, and key setup differences would substantially improve clarity.
- Clarify the conditional-generation setup. The paper uses a class-conditional ImageNet-pretrained LDM and CelebA-HQ gender labels. The authors should explain how the conditioning space is adapted from ImageNet labels to binary gender labels, whether any conditioning layers are reinitialized or trained, and whether those parameters are included in the reported trainable parameter counts and privacy accounting.
- Add visual samples for representative settings, especially ε ∈ {1, 5, 10, ∞}, different adapter placements, and rank choices.
- Improve writing and polish.

---

> ### Author Response · Authors · 2026-07-13
> **Author Response to Reviewer Comments**
>
> We thank the reviewer for the detailed and constructive feedback. We appreciate the suggestions and have carefully revised the paper to address the concerns raised. The revisions improve both the scope and clarity of the paper. Below we summarize the key changes.
>
> 1. Scope of reproducibility and robustness claims
>
>     We agreed that our initial version overstated the scope of our experiment. Accordingly, we revised the title, abstract, introduction, discussion, and conclusion to present our paper as a reproducibility study for a single practical backbone substitution setting, rather than generalizing our claims across architectures.
>
> 2. FID evaluation protocol and comparability
>
>     We expanded the methodology to clarify our evaluation protocol as compared to the original paper. The revised paper now specifies the FID sample size difference, reference dataset, and other principal differences. We added this information as a concise, easy-to-read table. Additionally, we explicitly state the effect of our choices (increased variance), and hence, motivate our focus on evaluating qualitatively instead of strictly numerically.
>
> 3. DP accounting
>
>     We added a dedicated table in the appendix describing the DP configuration, including details like optimizer, accountant, Opacus version, sampling rate, etc. We also clarified that the class-conditional embedding is trainable and included in the DP-SGD accounting.
>
> 4. Conditional generation setup
>
>     We expanded the methodology to explain how the ImageNet-pretrained model is adapted to the binary gender labels required in CelebA-HQ.
>
> 5. Localized DP experiment
>
>     We agreed that the original presentation could be interpreted too broadly. References to RoI-DP have been revised and framed consistently as a feasibility study. We state that only the selected parameter subset receives a formal DP guarantee and discuss the underlying assumptions and limitations. Additional details are available in the Appendix C.
>
> 6. Comparison with original work
>
>     To strengthen the reproducibility aspect, we added a direct comparison between the qualitative findings as reported in DP-LoRA, and the observations reproduced in our study, insted of a numerical comparison. The table in the beginning of section 5 helps motivate why we opted for qualitative comparisons instead of Absolute FIDs.
>
> 7. Broader impact, limitations, and paper format
>
>     We added a dedicated Broader impact section in the discussion and reorganized our paper. Details that do not directly support our results have been moved to the appendix (hyperparameter table, model configurations). We also explicitly discuss limitations regarding single-seed evaluation and implications of these choices for reproducibility.
>
> 8. Qualitative image samples
>
>     We appreciate the suggestion to included representative generated samples and investigated this during revision. However, because the original pretrained checkpoints are unavailable and our substituted pretrained backbone produces noticeably different visual characteristics, we believe qualitative image comparisons could be misleading and confound the scientific comparison. Since the original work primarily evaluates using FID, we choose to focus on qualitative trend-based comparison while discussing the limitations due to our backbone substitution.
>
>
> We thank the reviewer once again for their constructive feedback. We believe the revised paper addresses the principal concerns regarding scope, methodology, DP accounting, and presentation, while providing a clearer and more transparent reproducibility study.

---

### Review · Reviewer_BJza · 2026-07-14

**Summary Of Contributions:**

The paper reproduces the results of the work “Differentially private fine-tuning of diffusion models” by Tsai et al. ICCV 2025. In particular, they fine-tune a pretrained diffusion model on CelebA dataset with DP-LoRA in various training settings, confirming the findings of the original work. The authors also extend the evaluation to the “localized differential privacy”, where only parts of the images are considered to be private. This notion, not guaranteeing a formal protection, allows to make the models more accurate, while empirically maintaining some privacy.

**Audience:**

No

**Audience Explanation:**

From the manuscript it is not clear what is the motivation to reproduce the work of Tsai et al., why this specific work?

It was omitted by the authors that the DPImageBench [1] also tests the method of Tsai et al. for multiple datasets. If the main motivation was to verify if DP-LoRA for diffusion models would work in other settings, besides those demonstrated in the original paper, I believe DPImageBench has already provided a positive answer.

It would be more valuable if the authors would reimplement the method of Tsai et al. rather than relying on their code. If there are bugs and mistakes in the original implementation, it could be caught by the reimplementation, giving more confidence in their method.

[1] Gong et al. “DPImageBench: A Unified Benchmark for Differentially Private Image Synthesis” CCS 2025

**Broader Impact Concerns:**

No concerns

**Claims And Evidence:**

Yes

**Claims Explanation:**

The authors describe their evaluation in a fair amount of detail and provide the code, which is mostly based on the original work of Tsai. The results support the findings of the original paper in the computationally tractable regimes. The newly introduced localized differential privacy experiments look reasonable and are explained sufficiently.

**Requested Changes:**

It would be better for the paper to reimplement the procedure of Tsai et al., rather than relying on their code, in this way one could additionally verify whether mistakes were made in the original implementation.

I would suggest to include more details for the motivation of the paper, in particular, I do not understand why the paper of Tsai et al. had to receive a reproducibility study. And why the work of DPImageBench has not already provided sufficient evidence towards reproducibility.

**Minor changes.**

Most of the references in the paper are included without the venue, where they are published, even including the work of Tsai et al. itself.

---

> ### Author Response · Authors · 2026-07-20
> **Author Response to Reviewer Comments**
>
> We thank the reviewer for their constructive feedback. We appreciate the comments regarding the motivation of this study, its relationship to existing reproducibility efforts, and the role of reimplementation. We have revised our paper to some extent.
>
> 1. Motivation for reproducibility study
>
>     We expanded the introduction and Scope of Reproducibility sections to better motivate why DP-LoRA is an important target for reproduction. DP-LoRA is a representative parameter-efficient approach for differentially private diffusion model fine-tuning, yet reproducing its findings is challenging because of the unavailability of the original pretrained checkpoints. Consequently, independent researchers must substitute the pretrained backbone, making it unclear whether the reported qualitative trends persist under this practical setting. Our work specifically investigates this reproducibility question.
>
> 2. Relation to DPImageBench
>
>     Thank you for pointing out DPImageBench. We clarify its relationship to our study in the Related Work section. While DPImageBench benchmarks DP-LoRA alongside other private iamge sythesis methods across multiple datasets, it does not study whether the qualitative findings of the original paper remain reproducible when the original pretrained backbone is unavailable. We therefore view the two works as complementary rather than overlapping.
>
> 3. Use of the original implementation
>
>     We agree that an independent reimplementation can provide additional verification and may reveal implementation errors. However, the objective of our work was to evaluate reproducibility under backbone substitution while minimizing any confounding factors. Using the authors' released implementation allows observed differences to be attributed primarily to the substituted pretrained backbone and experimental setup rather than to implementation differences. We document all modifications made to the released implementation and make them publicly available for transparency.
>
> 4. Reference formatting
>
>     We updated the bibliography to include publication venues where applicable and corrected the formatting of the references throughout the paper.
>
> We thank the reviewer again for their suggestions. We believe our revisions better motivate the paper, its relationship to existing work, and improve the presentation of our reproducibility study.

---

### Review · Reviewer_aZcs · 2026-07-16

**Summary Of Contributions:**

This paper studies reproducibility of DP-LoRA (Tsai et al., 2024) for differentially private fine-tuning of latent diffusion models using a backbone substitution. They find that DP-LoRA trends remain stable for the substituted backbone.

**Strengths:**

- The paper studies whether DP-LoRA’s findings still hold when the original checkpoints are unavailable and a public backbone must be substituted, which is practically meaningful.

- The experiments support the main privacy–utility trade-off, the ablation trend, and the parameter efficiency of DP-LoRA.

- Code is available in the anonymous repository link.

**Weaknesses:**

- Only one substituted backbone is tested. The study shows that the trends hold in one practical setup, but it does not establish that they generalize across different backbones.

- The main experiment is conducted only on CelebA-HQ, so it is unclear whether the findings generalize to other image domains, for example medical datasets.

- Localized DP does not provide full-model privacy.

- Experiments use only one random seed.

- I recommend adding qualitative image examples of DP-LoRA to show the visual quality.

- The paper does not evaluate on membership inference, reconstruction, or other real privacy attacks.

**Audience:**

Yes

**Audience Explanation:**

DP-LoRA is a useful approach for adapting large diffusion models to sensitive data. Researchers working on differential privacy and diffusion models may be interested in knowing whether its main trends remain reproducible when the original checkpoints are unavailable.

**Broader Impact Concerns:**

No.

**Claims And Evidence:**

No

**Claims Explanation:**

The main study uses only one substituted backbone and one dataset. This makes it difficult to know whether the reported qualitative trends generalize beyond this specific setup. In addition, the paper does not clearly justify why the cin256 backbone was selected over other publicly available alternatives, beyond its availability.

**Requested Changes:**

- Evaluate more than one substituted backbone, and provide a clearer justification for selecting them.

- Include additional datasets in different image domains.

- Add qualitative image examples.

---

> ### Author Response · Authors · 2026-07-20
> **Author Response to Reviewer Comments**
>
> We thank the reviewer for their constructive feedback and positive assessment of the practical motivation for our work. We have revised the paper to better reflect the intended scope and limitations of the study.
>
> 1. Scope of backbone substitution
>
>     We agree that evaluating a single substituted backbone does not establish general robustness across architectures. Accordingly, we have revised our title, abstract, introduction, discussion, and conclusion to consistently present our work as a reproducibility study for one practical backbone-substitution setting rather than making broader robustness claims. We also expanded the methodology to justify the selection of cin256 based on its public availability and architectural compatibility with the original DP-LoRA implementation.
>
> 2. Dataset scope and experimental limitations
>
>     We acknowledge that evaluation on additional datasets and multiple random seeds would strengthen the conclusions. Due to computational constraints, our experiments remain limited to CelebA-HQ and a single seed. We now discuss these limitations explicitly in the paper and frame our conclusions accordingly.
>
> 3. Localized DP
>
>     We have revised the presentation of the Localized DP experiment throughout the paper. It is now consistently described as a feasibility study rather than a complete differential privacy mechanism, and we clarify that only the selected parameter subset receives a formal DP guarantee while discussing the underlying assumptions and limitations.
>
> 4. Qualitative image samples
>
>     We appreciate the suggestion to include representative generated images and investigated this during revision. However, because the original pretrained checkpoints are unavailable and our substituted pretrained backbone produces noticeably different visual characteristics, direct visual comparisons with the original work could be misleading. Since the original paper primarily evaluates using FID, we instead focus on reproducing the reported qualitative trends while explicitly discussing this limitation.
>
> 5. Privacy attacks
>
>     We agree that evaluating membership inference or reconstruction attacks would provide valuable additional evidence. Such experiments were beyond the scope of our reproducibility study, whose objective is to reproduce and extend the experiments reported in the original paper. We identify empirical privacy attacks as an important direction for future work.
>
> We thank the reviewer once again for the constructive comments. We believe the revised manuscript more clearly defines the scope of our study while improving the presentation of its assumptions, limitations, and contributions.